# The Status of *Prunus padus* L. (Bird Cherry) in Forest Communities throughout Europe and Asia

**Rolf D. J. Nestby**

The Norwegian Institute of Bioeconomy (Horticulture), 1431 Ås, Norway; rolf.nestby@getmail.no;
Tel.: +47-95988530

**Abstract:** *Prunus padus* L. (bird cherry) belongs to the *Racemosa* group in subgenus *Padus* in the genus *Prunus* L. It is a hardy invasive species, which makes it valuable for securing slopes, and for eco-design. It is a good solitary park tree with early flowering of white flowers in racemes, which have a pleasant smell. However, it may be attacked by cherry-oat aphid, and the small ermine moth, which may weave giant webs over the whole tree, which demonstrates the important role of *P. padus* in the food web of forest ecosystems. The species is in balance with these pests, other herbivores and diseases throughout Europe and Asia. Another threat is the competition against the invasive *P. serotina*, but it seems that *P. padus* is not strongly threatened, though they compete for the same habitats. Moreover, human interference of forest community ecology is probably the greatest threat. The tree is not only winter hardy; it can also survive hot summers and tolerate a wide variety of soil types. It may form dense thickets due to the regeneration of branches bent to the ground and basal shoots, and may be invasive. These characteristics are important in determining the ecological niche of *P. padus*, which involves the position of the species within an ecosystem, comprising both its habitat requirements and the functional role. It is also important that *P. padus* has effective dispersal of pollen and seeds. This, together with the previously noted characteristics and the fact that the tree can cope well with climate change, define it as a not threatened species. However, the ssp. *borealis* is threatened and national level monitoring is required. *Prunus padus* has been exploited by farmers and rural population, but is less used today. However, it is still used for making syrup, jam and liquor. Moreover, the wood is valuable for wood carving and making cabinets. All tissues are valuable as sources of powerful natural antioxidants. However, the interest in the *P. padus* fruit and other tissues is overshadowed by the interest in other wild species of edible and human health-related berries. Moreover, the tree is used in horticulture as an ornamental in gardens and parks, values that deserve a new focus.

**Keywords:** botanical classification; community ecology; phenology; herbivores; human interference

## 1. Introduction

Phenology is the study of natural biological events in relation to climate. The registration of phenology data in Europe goes back to early 1700s and, and these are probably the oldest biological data of their kind. These data show that the experienced climate changes have caused the winter to be squeezed at both ends, making the growing season longer [1]. This also counts for *Prunus padus* L., which is the most widely distributed of the *Prunus* species and extends throughout Europe from northernmost Scandinavia and northern Russia to the mountains of the Iberian peninsula and has been recorded in Morocco. It also occurs in northwestern Italy, Croatia, Bulgaria and northeastern Asia-minor, extending eastward into Asia from W. Siberia as far as Caucasus and the Himalayas, northern China and Japan (Figure 1). It is also established as an invasive alien in North America (Alaska). It mainly thrives in wet woodland, hedgerows and on river banks and is very hardy

but vulnerable to drought. Trunks 80 cm in diameter have been reported from the west coast of Norway [2–9]. The focus of this article is to give a status of the biological classification and phenology of *P. padus*, its position as a valuable tree in the forest community ecology and to manifest its ornamental and human health-related value.

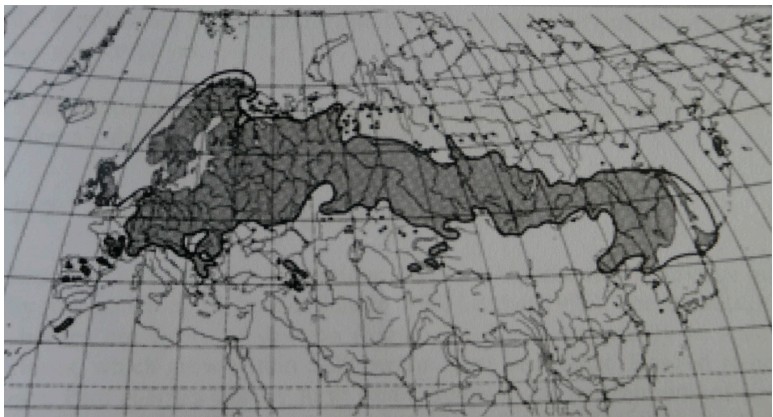

**Figure 1.** Distribution of *Prunus padus* L. in Europe and Asia [2].

## 2. Botanical Classification and Phenology

### 2.1. Classification

*P. padus* is a tree or a bush (2–14 m) which develops a stone fruit, black in color when ripe. The species belongs to subgenus *Padus* in the genus *Prunus* L. (*Rosacea*) (Figure 2). The polyphyly of *Prunus* subgenus *Padus* and position within the *Padus* group has been thoroughly studied, but it is complicated. In 2016, *P. padus* L. was placed in the polyploid *Racemosa* group in subgenus *Padus* [10–12].

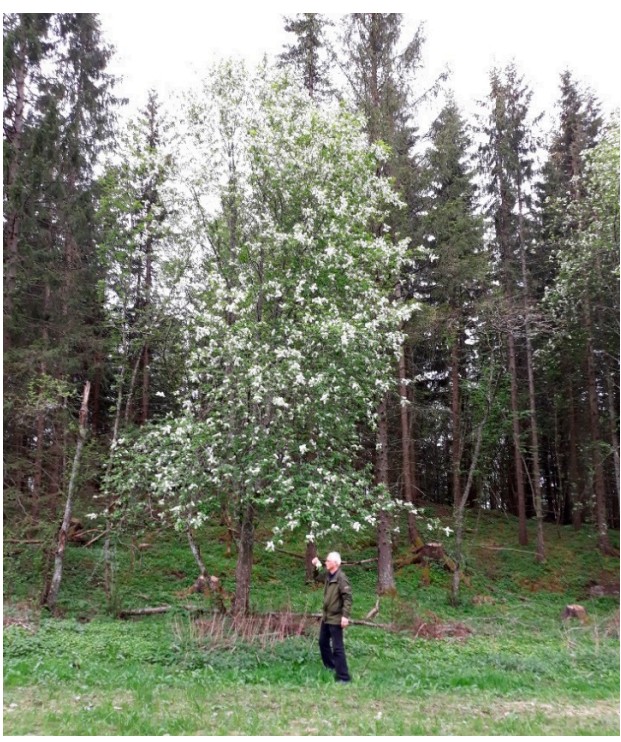

**Figure 2.** Flowering solitary tree of *Prunus padus* L. at a forest edge in Stjørdal municipality in mid-Norway, 21 May 2019. The tree is 10–12 m high and the trunk diameter at chest height is 29 cm. Photo: Rolf Nestby.

The genus *Prunus* is generally divided into two sub-species (ssp.): *P. padus* ssp. *padus* L. (European bird cherry), which is a small tree, and *P. padus* ssp. *borealis* L. (synonymous with *P. padus* ssp. *petraea*), which is a bush (2–4 m) and has shorthaired leaves. *P. padus* ssp. *padus* grows in riparian forests in the lowlands of Europe and Asia, while *P. padus* ssp. *borealis* grows in the arctic part of Scandinavia and in the alpine and subalpine regions of the Alps and in the Carpathian Mountains [6,9,13–16]. Almost 20 different cultivars or forms have been named, such as 'Watereri' with large flower racemes [6,9].

*2.2. Phenology*

The twigs of *P. padus ssp padus* are dull deep brown with pale markings. Its shoots are hairy when young but become hairless with age. The leaves are oval and hairless. The edges have fine, sharp serrations, with pointed tips and two glands on the stalk leaf base. The tree is a hermaphrodite. Bisexual flowers in racemes appear after the sprouting of leaves (April–June), and provide an early source of nectar and pollen for bees (Figure 3). The flowers are strongly scented, white and normally have five petals, and measure 6–15 mm across and are protogynous, while subspecies *borealis* has leaves with short hairs, the flower clusters are generally upright and few, and the flowers have no smell [3,6,9,13,14].

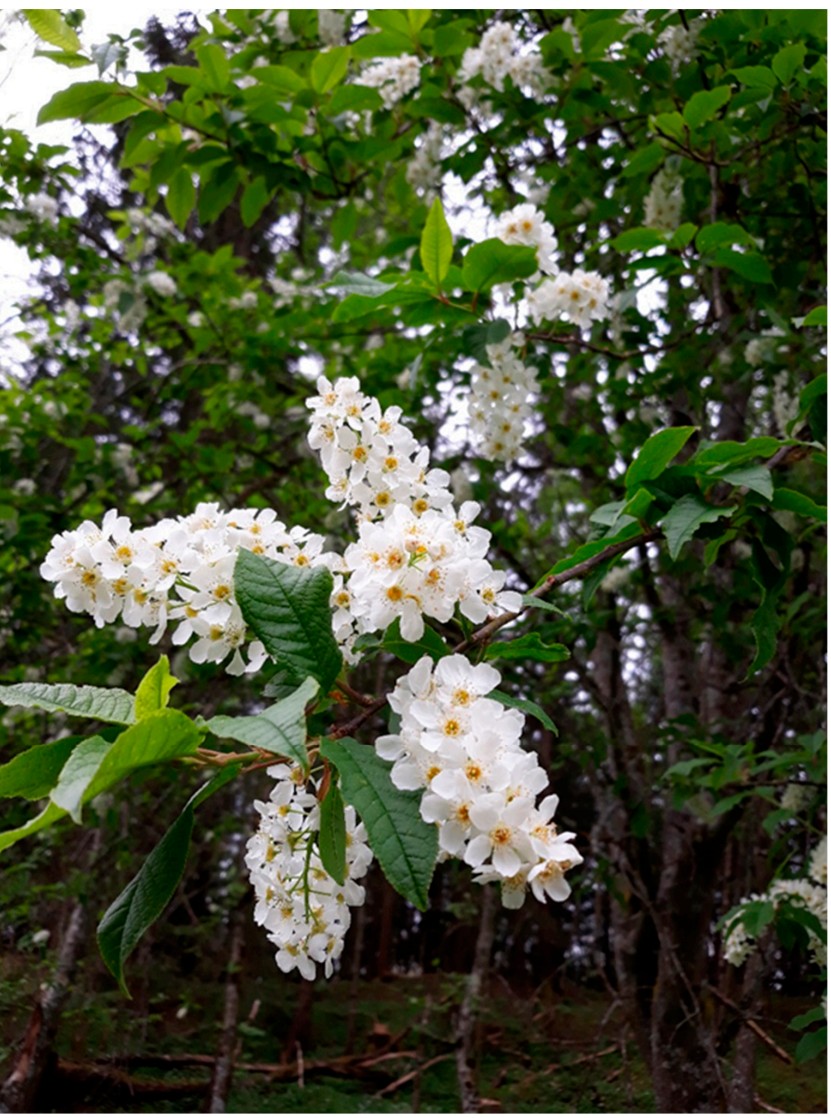

**Figure 3.** Racemes, leaves and shoots of *Prunus padus* ssp *padus* L. with open flowers. Photo: R. Nestby.

*Prunus padus* is a hardy species and is the most northerly distributed *Prunus* species in Europe, and the northerly distribution limits broadly coincide with the shores of the Arctic Ocean. In the Alps, it grows at a higher altitude than any other deciduous tree [4,6]. Parallel to this, there is strong evidence that *P. padus* persisted in northern Europe during the last glacial maximum (approx. 20,000 Before Present, BP), based on the presence of the co-dependent leaf beetle (*Gonioctena internodia*) in northern Europe since the beginning of the last glaciation [17].

Based on historical data, using a simple degree-day model, the average base temperature for the unfolding of leaves is 3.2 °C, and for the start of flowering it is 3.4 °C [18]. The onset of flowering is correlated with the North Atlantic oscillation index of the proceeding winter. Flowering occurred earlier following positive winter conditions in nearly all early flowering species, among them *P. padus* [19]. Climate warming may increase the relative importance of chilling in the timing of bud burst, which should be noticed when using phenology models to calculate spring events [20]. Concerning climate, *P. padus* is an important species when it comes to carbon storage, with an average value of 47.6% in the biomass as carbon [21].

Seed Dispersal

For sessile plants (like *P. padus*), dispersal poses a serious problem, because they depend on wind or animals to spread their pollen and seed. For the spread of pollen, the observation of above 75 percent decline in total flying insects in protected areas in Germany is alarming [22]. It could reduce the fruit set in all species dependent on insects for pollination. Once flowers are pollinated and fruits are set, the seeds have to be dispersed. For example, short distance seed dispersal of *Prunus mahaleb* was mainly facilitated by passerine birds, with larger birds and mammals removing substantially fewer fruits. However, more than half of the seeds eaten by passerines were dispersed within 50 m of their maternal parent, whereas larger birds dispersed seeds more than 110 m, and foxes over distances more than 500 m [23]. The Eurasian Jay (*Garrulus glandarius*) is an important agent of zoochory and participant of the succession processes in European forests. It can transfer seeds of *P. padus* over a distance of 900–1100 m, which corresponds to the radius of the summer home range of the jay [24]. The loss of an effective seed disperser, such as Asian black bears, did not affect the spatial genetic structure (SGS) in wild cherries, probably because of compensation of the bear loss by other vertebrate dispersers and too few tree generations after the bear loss to alter SGS [25]. *P. padus* is closely related to *P. mahaleb* and its seed dispersal could be expected to have a similar pattern. However, because *P. mahaleb* and *P. padus* establish and grow better under the protection of shrubs than at open sites where the mammals often disperse the seeds, the many long distance dispersal events therefore do not necessarily guarantee successful recruitment, although one successful immigrant per generation is sufficient [9,26].

In horticulture, birds feeding on cherries are a serious problem. In Michigan, USA, observation studies showed that chipping sparrows (*Spizella passerine*) were the most commonly detected species on tart cherries (*P. cerasus*), while other bird species preferred sweet cherries (*P. avium*). [27]. Thus, not all bird species in Europe and Asia are likely to feed on *P. padus* (because of tart fruits), which is of interest for understanding the dispersal effectivity of the species. There is a weak selection gradient favoring large seeds, but this selection gradient is not clearly related to habitat [28].

A study on seed dispersal showed that wild cherry trees (*Prunus avium*) are surprisingly robust to changes in habitat and landscape structure. The movement behavior of birds, and possibly adaptive increases in movement distances in structurally poorer landscapes, lead to high seed removal rates and potentially directed dispersal of cherry seeds to suitable habitat patches. Such processes might also be common in other systems where ecological function depends on highly mobile organisms, such as pollination systems and interactions between parasites and parasitoids and their hosts [29]. Moreover, the spatial distribution of *P. padus* and maple (*Acer platanoides* L.) pollination was examined in Estonia in a time series from 1948 to 1995. It showed that rate of spread (timing) of phenology phases was very different in early and late years. In early years, the phenology phase moved very fast, except for

the western islands. The onset of maple phases is gradually directed more towards the northeast and *P. padus* more to the milder climate of the northwest [30].

## 3. Community Ecology

### 3.1. Species Richness and Canopy Height

The interactions between forest communities and ecosystems are reflected in species richness and abundance patterns [31]. A community is affected by strong interspecific competition at small spatial scales. This type of competition improves species diversity by promoting niche differentiation, and though the importance of neutral processes increases with spatial scale, the contributions of niche processes cannot be ignored [32,33]. This is of interest concerning the niche of *P. padus,* which is a small tree or a bush. In a specification of understory aged ≥ 10 years, as well as its species composition in the biomass of the Niepolomice forest (Southern Poland, Near Krakow), the most common species was *Frangula alnus* (56.8% of the plots). Quite common ones included *Sorbus aucuparia* (27.7%) and *Prunus padus* (12.8%) [34]. A global correlation between vascular plant species richness and average forest canopy height demonstrated a significant correlation both at the global and macro-climate scales, suggesting that the volume of forest ecosystems should be considered in ecological studies as well as in planning and managing natural sites. High-resolution spatial data could be highly important to confirm the relation between species richness and canopy height, even at different scales [35].

### 3.2. Phenological Influence

*P. padus* is not only winter hardy; the tree can also survive hot summers and tolerate a wide variety of soil types. It may form dense thickets due to the regeneration of branches bent to the ground and basal shoots, and may be invasive [6,9]. These characteristics are important in determining the ecological niche of *P. padus*, which involves the position of the species within an ecosystem, comprising both its habitat requirements and the functional role [36].

The invasive character not only allows the species to create an ecological niche in its natural habitat, but also to spread as an alien along urban Alaskan salmon streams. Prey subsidies to urban Alaskan salmon streams from terrestrial invertebrates (e.g., caterpillars) present on the foliage of *P. padus* demonstrated no ecological effect on the coho salmon (*Onchorhynchus kisutch*) (these subsidies were lower than for native deciduous trees). However, reductions in subsidies are likely to have a negative consequence for the salmon as *P. padus* continues to spread [8].

In a study of the structure mechanism of tree species diversity pattern (how trees are spread or scattered) in a near-mature forest in China, *Prunus padus* was at a 0–50 m scale (distance from the tree trunk), together with several other tree species classified among diversity accumulators [37]. This implies that *P. padus* has positive facilitative interactions (easily forward interaction with other species) with other species, and that the target species would accumulate and maintain an over-representative proportion of diversity in its proximity (nearness in place) [38].

#### 3.2.1. The Herbivore Influence

Some herbivores create more problems for *Prunus padus* than others, namely the cherry-oat aphid *Rhopalosiphum padi* L., and the small ermine moth *Yponomeuta evonymellus* L., and to some degree the leaf beetle *Gonioctena quinquepunctata* Fabricius. These are generally spread in Eurasia. However, other herbivores may cause regional problems.

#### 3.2.2. The Cherry-Oat Aphid (*Rhopalosiphum padi* L.)

*P. padus* is a winter host of the cherry-oat aphid which in spring migrates to species of the grass family. When testing the influence of neighbor plants and drought, inter-varietal interaction was shown in cherry-oat aphid feeding on wheat (*Tritium aestivum* L.), one of the possible summer hosts of the aphid. The winter host plant specificity of *R. padi* is controlled mainly by the preference of

the female remigration to the winter host. The second spring generation, if winged, migrates to the summer host, grass [39,40]. Over-expression of the aphid-induced serine protease inhibitor C12c gene in barley affected the generalist green peach aphid, but not the specialist bird cherry-oat aphid [41], and plant to plant interactions reduced aphid performance and generated associational resistance. However, in diverse mixtures of plants, drought stress greatly diminished the genotypic diversity driven reduction in aphid performance [42].

Another question is if *P. padus* has some means to reduce the stress caused by the aphid. It is shown that the infestation of the aphid induces enzymatic activities [ornithine decarboxylase, lysine decarboxylase (LDC) and tyrosine decarboxylase] within tissues of less susceptible wheat, and this tendency was the strongest for LDC. This activity induces the production of compounds (amids) that block synaptic transmission through binding to quisqualate-type glutamate receptors on exosceletal muscles of arthropods. Free exogenous polyamids may also disturb the feeding, survival and settling behavior of cereal aphids [43]. Moreover, a study on the interaction between bird cherry-oat aphid (*R. padi*), the ladybird (*Coccinella septempunctata*) and the parasitoid *Aphidius coliemani* suggested that airborne interaction between undamaged plants of barley cultivars can affect insects at higher trophic levels, and that odor differences between different genotypes of the same plant species may be sufficient to affect natural enemy behavior [44], strengthening the warning of using pesticides in these habitats. Pesticides would have a negative effect on the natural balance, and thereby interfere with the natural balance of species which have developed through thousands of years [45].

### 3.2.3. The Small Ermine Moth (*Yponomeuta evonymellus* L.) and the Leaf Beetle (*Gonioctena quinquepunctata* Fabricius)

Another important herbivore on *P. padus* is the small ermine moth (*Yponomeuta evonymellus* L.), considered an obligatory monophagous insect pest that originally feeds only on native *P. padus* L. The caterpillars may weave giant webs over the whole trees, including the trunks [7,46]. However, increasing larval feeding on alien *P. serotina* Ehrh. has been observed, with the leaves of the shrub species growing in full sunlight being less injured than those in shade, probably due to higher concentrations of defense metabolites and lower concentrations of N [47]. The effect of the host species on light conditions was not significantly related to feeding on leaves with subsequent defoliation. However, mass fecundity in all the studied wing parameters were in contrast higher in larvae that grazed on *P. padus* than on *P. serotina*. Similarly, the same parameters were also higher on shrubs in high light compared to low light conditions. In general, light conditions, rather than plant species, were more often and to a greater extent responsible for differences in the observed parameters of insect development and potential fecundity [48–50]. The identity of the host plant species during larval feeding determines a preference to deposit or lay eggs (oviposition) on that host species. The larvae grow equally well on leaves of the *P. serotina* species as on *P. padus*, and therefore the reason that feeding of larvae on *P. serotina* in nature is low compared to *P. padus* is mainly due to phenology-related reasons [47,51]. A recent study did not support the hypothesis that larval feeding of *Y. evonymella* on *P. serotina* solely is to avoid parasitoids and predators, and therefore did not appear to be associated with the enemy-free space hypothesis. Most likely, the main benefit for *Y. evonymella* is the expansion of its food base [52].

The fluctuation pattern in *Y. evonymella* feeding on leaves of *P. padus* could not be explained by fluctuations in berry production, as is the case with *Paraschwammerdamia lutarea* feeding on leaves of *Sorbus aucuparia* [53]. Moreover, synchrony in abundance of population variation comparing two areas (habitats) in Finland remained at short and intermediate distances; the magnitude of peak defoliations recorded differed between locations, and the magnitude of the population variations of *Y. evonymella* was unequal. One factor is the negative relationship between pupal mass and current year tree defoliation, which suggests that shortage of food resources affected the reproductive potential of *Y. evonymella* during the high peak defoliation years. This was not observed in the low peak area. Moreover, parasitism was a factor with influence on population size depending on the

degree of defoliation [54]. Among the parasitoids, *Diadegma armillatum*, *Herpestomus brunnicornis* and *Zenilla dolosa* were the most important with 3.5%, 7.1% and 7.7% of the combined parasitism of the host larvae and pupae, respectively [55]. It seems that as long as there are no human or other interferences (such as catastrophes) in the ecological communities weakening the position of *P. padus*, it will survive the herbivore-created stress.

It should also be mentioned that the leaf beetle *Gonioctena quinquepunctata* may be a serious leaf pest on *Prunus.* Interactions between phenological factors, the availability of good food and more factors may explain the pattern of growth and development of the polyphagous *G. quinquepunctata* on leaves of the two *Prunus* species *P. padus* and *P. serotina*. The leaves are acceptable food for both larvae and adult beetles of these species. However, under conditions in which multiple plant species thrive, this insect prefers native species, such as *P. padus* or *Sorbus aucuparia* L., over the alien *P. serotina* or other woody plant species [56].

### 3.2.4. Injury by Other Pests

Locally, other pests may be a problem. In eastern Fennoscandia, *P. padus* is the main host of *Operophtera brumata* (the winter moth). In the 1990s, an outbreak of the moth resulted in a large reduction in radial growth of the trunk of *P. padus*. However, less preferred tree species did not benefit from the outbreak [57]. Another problem are the gall mites *Eriophyes padi* and *Aculops padus* Xie, which were observed on *P. padus in* Slovakia, Germany, Ireland, France and China, where they formed gals on the backside of the leaves [58–62]. Moreover, *P. padus* is susceptible to bacterial cankers caused by *Xanthomonas arboricola* pv *pruni,* and *Pseudomonas syringae* [14,63,64]. Further, *Prunus padus* is a host of cone rust in Norway spruce caused by the rust fungi *Thekopsora areolata,* that also can kill the top shoots of Norway spruce, and is heteroecious with *Chrysomyxa pirolata* that forms spermatogonia (undifferentiated male germ cells) [65,66]. Also, viruses like Prune dwarf virus and Prunus necrotic ring spot virus may be a problem in *P. padus* [67]. It has to be mentioned that none of the wild *Prunus* (among them *P. padus*) tested positive for the peach latent mosaic viroid in a Croatian survey [68].

### 3.2.5. Injury Caused by Mammals

The preference of feeding of the European beaver (*Castor fiber*) on different tree species varied between locations. However, *P. padus* was among the preferred species [69]. Moreover, rooting by wild boar (*Sus scrofa scrofa* L.) in central Netherlands had no negative effect on *P padus*, *P. serotina* and several other tree species. However, oak (*Quercus robur* and *Q. petraea*), red oak (*Q. rubra*) and beech (*Fagus sylvatica*) were negatively correlated with rooting frequency [70], which would disfavor their establishment in the forest, and probably be a benefit for *P. padus*.

### 3.2.6. Stress Mechanisms against Herbivory

To fight herbivory, *P. padus* has different mechanisms that are important but costly. Theory predicts that resources are allocated for defense to optimize the investment. For a tolerant competitor such as *P. padus*, this has obvious benefits since it does not have to maintain the defensive secondary metabolism, and can instead direct its energy to growth, which in itself improves the competitiveness [71]. Moreover, tree defoliated by heavy feeding of the small ermine moth allocated their resources to defense during the most intensive growth [72–76]. Furthermore, when the enzymes of herbivores decompose cell walls, chemical signals are induced and carried through the cytoplasm and the plant xylem. Intact cells which have an active metabolism will start producing phytoalexins after a couple of hours. If the cell structure is damaged, enzymes, glucosides, and mobile toxic compounds will be released, as well as complex polyphenols from sugar parts, and accumulate in damaged parts [77]. It is argued that with induced resistance, plants can repel the feeders, which arrive later in the season. By this mechanism, new and remaining foliage of the tree will produce enough assimilates for storage, which in return are used for growth or defense the following year [72].

*3.3. Human Interference on Forest Ecology*

　　Bringing tree species from other continents into Europe/Asia is an example of human interference. The already mentioned alien invasive *P. serotina* is a competitor to *P. padus*, and thus represents a threat. A study in Central Poland proved that effects of propagule sources were strongly modified by habitat features, which confirms that plantation of conifers on sites suited for deciduous forests increases the risk of *P. serotina* invasion, and soil and light parameters seem to facilitate the invasion of *P. serotina*, and comprise the consequence of this process. Moreover, it may be assumed that the ecological success is connected with stochastic (random) processes in populations of the invader's seedlings rather than with stochastic release from competition caused by disturbances [78–81]. Another study in Poland showed that despite the high seedling density of invasive species beneath parental tree canopies, the survival was lower than in the surrounding forest. These two aspects of the establishment process contribute to the dynamic equilibrium between limiting and facilitating the growth of young generations of invasive plants [82]. Moreover, evolutionary processes may generate a specialized herbivore community on an invasive plant, allowing for reduced invasiveness over time. This suggests that the manual control of herbivores should be avoided, since it might have an adverse effect of a slowing down processes of adaption, and a thereby delay the decline of the invasive character of *P. serotina* [45]. It is interesting that the spring generations of the aphid *R. padi* do not survive on *P. serotina*, because aphids feeding on *P. serotina* had a considerable delay between finding and accepting the phloem [83]. However, phloem contact does not appear to be a prerequisite for these aphid forms to initiate reproduction [84]. The movement onto secondary hosts may be induced by changes in levels of free amino acids within host tissues, affected by long term aphid feeding, which evokes the proteolytic machinery, which in turn stimulates rejection by the primary host and migration onto secondary plant hosts [85,86]. To delay the spread of *P. serotina,* use of tree stand description as a model to assess the prevalence of *P. serotina* is suggested to replace laborious and expensive tests [87].

　　Rowan (*S. aucuparia* L.) is increasing considerably in abundance in Finnish urban forests (dominated by Norway spruce—*Picea abies* L.). However, a decrease in abundance of birch (*B. pubescens*) and an increase in a broad-leaved group (including *P. padus*) coincided with a decreasing number of rowans, and decrease in the basal area of Norwegian spruce. The authors concluded that regulating the tree species is not an easy way to keep rowan thickets under control [88].

　　In line with this, it was found that fears concerning a negative impact of meadows on natural forest vegetation through the penetration of alien plant species and phytophagous *Chloropidae* (grass flies) seem to be unfounded. Ecotone (such as a meadow) was an important barrier for most plant and *Chloropidae* species in their dispersion from meadow to forest and vice versa [89]. The ecotone effect could reduce the pressure from *P. serotina* on *P. padus* of man-made constructions, like electric pylons, which make perching sites for birds. Under these, the density of flesh-fruited species was higher than in control plots, and 85% of the fleshy-fruited species were represented by alien species. Among these, *P. serotina* was one of the species most frequently found [90]. A "positive" character of *P. serotina* is that it is a new food plant of caterpillars of the Scarce swallowtail (*Iphiclides podalirius*), which is considered a valuable butterfly (e.g., a good pollinator) [91].

　　Natural regeneration following soil scarification in forest microsites in Estonia, a method also used in Norway and in other parts of Europe (authors comment), consistently increased the number of conifer seedlings in the sites typical of Norway spruce (*Picea abies* L.) and Scots pine (*Pinus sylvestris* L.), and no suppressive effect was detected in the number of deciduous saplings (among them *P. padus*) [92].

　　A comparison between *P. padus* and *P. serotina* showed that *P. padus* accumulated a higher amount of particulate matter (PM, airborn particles that settle on the leaves) on the surface of the leaves. A strong negative correlation was found between the amount of PM accumulation and the efficiency of the photosynthetic apparatus in *P. padus*, but not in *P. serotina*. [93].

　　The ongoing change of climate will affect forest dynamics, but this effect may be slow, and will probably be overruled by catastrophes, such as the invasion of forest pathogens; e.g., invasion of Dutch

elm disease with the following reduction in the elm populations, which lead to an increase in saplings of other tree species, among them *P. padus* [94].

Another factor that could influence the competition and balance between tree species in a forest is anthropogenic pollution. Habitat rehabilitation experiments over five years in the Czech Republic showed that a human-disturbed river influenced a landscape with absence of natural vegetation; mulching the soil with straw was effective in restoring native tree species, among them *P. padus* [95].

## 4. Forest Community and Ecosystems Interaction

*Prunus padus* grows generally in a typical riperian forest, often together with *Alnus incana*. What characterizes this forest type is the interaction of species-specific life history strategies (i.e., reproductive mechanisms and physiological adaptions) with physical processes that result in the characteristic patterns of vegetation colonization, establishment and succession in a montane region, exhibiting characteristic vegetation patterns in response to similar climate, physiography, plant genera distribution, and disturbance regimes. This forest type is connected to river basins and generally to wetland, and also includes the hillsides connected to the river basin [96]. In such a forest type, comparing regeneration mechanisms of woody species in the hardwood floodplain forest of the Upper Rhine (Germany), it appeared that many woody species have developed strategies favoring vegetative propagation for their regeneration. This concerns understory species such as *Cornus sanguineae* and *Prunus padus* in particular. These species might be advantaged by regular and prolonged flooding of the Rhine forest over species regenerating only by seeds [97]. Parallel to this, in grey alder (*Alnus incana*) stands, the species that most clearly describes (a differentiating species) the Croatian association *Equiseto hyemali–Alnetum incanae* Moor 1958 is *Prunus padus.* This is a continental region along the watercourse of the river Drava. Here, the grey alder forest occurs mainly in riparian and floodplain forests [98]. In Bryansk Polesie, frequent cuttings have created a black alder (*Alnus glutinosa*) forest, which has become the dominant forest type in lowland swamps. This management has prevented the succession of the black alder forest into a tall herb spruce forest, which is close to a community of the climax maximum type. This tall herb spruce forest type also contains *P. padus*, among a wide spectra of tree species [99]. The frequent cutting (every 60–70 years) would probably have a negative effect on *P. padus* by forcing it to adapt to a different community type. In a study of a long-term succession in a former raised bog after intensive cutting and protection since 1884, new plant species mainly naturalized through garden immigration (e.g., *P. padus*). Later, in 2005, the vitality and growth of many trees in the bog declined, and some of the taller individuals began to sink into the peat layer. Thus, in the future, the half-open bog forest may show some natural dynamics, allowing the survival of bog species in a mosaic of changing habitats [100]. However, not all riparian forests contain *Prunus padus*. A Hedero-helics-Alneutum-glutinosae forest (Alnenion glutinoso-incanae, Alnion incanae) in the inner Italian alps dominated by black alder (*Alnus glutinosa*) was distinguished, inter alia, by the absence of species preferring wet soil, like *P. padus*: this plant community is considered as endangered [101].

The sub-species *P. padus ssp borealis/petraea*, which is a bush, is adapted to a colder climate and grows in montane forests in the arctic regions of Scandinavia and in sub-alpine areas of the Alps and the Carpathian Mountains. In the Alps, it prefers a dry to humid slope and a light coniferous forest with green alder (*Alnetum viridis*) [13,15].

In interior Mongolia, only three tree species are distributed in Hunshandak Sandland, China (*Ulmus pumila, Malus baccata, Prunus padus*). *M. baccata* and *P. padus* were more sensitive to high temperature and irradiance than *U. pumila*. This is probably the reason for the high distribution of *U. pumila*. *Malus baccata* and *P. padus* are adapted to the back slope of fixed dunes, because this micro-habitat is relatively cool and less irradiated than the slope facing south. The water use efficiency of *U. pumila* was lower than for *M. baccata* and *P. padus*, the reason why *U. pumila* did not form forests in the dry soil [102]. This observation is in line with a vegetation survey of a south- and north-facing valley slope in Taebaeksan Provincial Park 850–1380 MASL, where *P. padus* was among tree species observed only on the north facing slope, away from the irradiation of the sun [103]. An observation

parallel to this in 2002 was that *P. padus* grew well in a scree slope and on a gentle slope with less rocky soil, but not on a moderately rocky soil > 1600 MASL in China [104].

Evaluation of 23 Norwegian tree species suggested that twelve widely distributed species with generally effective dispersal of pollen and seeds were considered viable (*Pinus sylvestris*, *Picea abies*, *Juniperus communis*, *Betula pubescens*, *B. pendula*,*Alnus incana*, *A. glutinosa*, *Salix caprea*, *Populustremula*, *Corylus avellana*, *Sorbus aucuparia*, *Prunus padus*), and have, as such, no particular conservation needs. The effective seed dispersal of these species, as inferred from post-glacial migration rates, may be partly responsible for their generally early post-glacial appearance, and may, in combination with the wide ranges and relatively large evolutionary potential, indicate that viable species are best able to cope with climatic change [105].

## 5. Attributes of Human Value

*P. padus* is not only a tree species that has value as a species in the ecological community of forests. It has also been used by humans for several reasons, like making hand tools, furniture, carvings etc. out of the wood, and for human diet and health-related reasons, as the fruits could be a valuable natural source of bioactive compounds with high antioxidant properties, due to the contents of organic and phenolic acids, catechins, and a synergetic effect of vitamin C and flavonoids [4,7,9,14,106–110].

### 5.1. Value for Human Health

All tissues of *Prunus padus* L. have been known since the middle-ages for their medicinal and high health-promoting values, and in addition, the fruits are used as food. In Norway, pits are found in human settlements dating back to the stone age, and the fruit was used later mainly to make alcoholic beverages, to some extent as jam and juices, and in traditional medicine. Crushing the pits gives a taste of almonds. However, the pit contains hydrogen-cyanide (prussic acid) in bound form as amygdalin, and stomach problems after too high an intake of products with crushed pits have been reported [7,9].

These findings support the suggestion that *P. padus* fruits, flowers, leaves and bark may be valuable sources of powerful natural antioxidants for use in food, medicine, cosmetics and other fields currently processing antioxidants [111–113].

### 5.2. Contents of Biological Compounds Valuable for Human Health

Ascorbic acid content is relatively low in *P. padus* fruits (5.22 mg $100g^{-1}$ fresh weight (FW)), while total anthocyanin content is 74.35 mg cyanidin glycoside (CG) $100\ g^{-1}$ FW, and total phenolic content 640.16 mg gallic acid equivalents (GAE) $100\ g^{-1}$ FW [114]. The free radical scavenging rate of procyanidin and ascorbic acid, extracted from fruits, shows a rising trend with increased concentration, and the inoxidisability of procyanidin extracted from fruits is higher than that of ascorbic acid [115,116]. In a study of six wild edible fruits, *P. padus* had the second highest level of total phenolic content (634.80 ± 30.77 mg GAE $100\ g^{-1}$ FW). The total flavonoid content was the highest among the species (165.55 ± 8.57 mg QE (quercetin equivalents) $100g^{-1}$ FW), and so was the antioxidant activity (2.95 ± 0.15 mmol Trolox $100g^{-1}$ FW); the high levels of phenolics were confirmed and in addition they found that the anthocyanin content was at the same level as for *P. cerasus* and *P. avium* (142.05 ± 10.04 mg $100\ g^{-1}$ FW) [117]. When adding hot extract from *P. padus* to cultures of *Lentinus edodes* (*Agaricomycetes*), a wood decay fungus used for the production of biotechnologically useful enzymes, such as laccases and peroxidases, and drugs, it showed that the hot extract contained an inducer of Mn-peroxidase not present in cold extracts, which may provide a lead in the chemical characterization of the inducer [118].

### 5.3. Other Values

The wood is today considered to have low commercial interest [4,14]; e.g., in Norway it was valuable in traditional farming because the wood was hard and tough. Logs were taken care of, cleaved and placed in the barn for drying. The wood was used as spikes in wooden rakes, as tholepins and generally as shafts of agricultural hand tools. Moreover, the bark was used for the dyeing of wool,

giving a yellow or green color, and as leather-tan [7]. Today, in our industrialized society, these values are not so important; however, the wood is still highly valued for wood carving in Norway (author's comment), and for making cabinets [4]. The value as fire wood is equal to birch (*Betula pubescens* L.) [119]. The invasive character of the tree makes it suitable for soil bioengineering to increase the stability of slopes and mitigate erosion, and it is suitable for shelterbelts and sound-breaks [3,4].

It could also be valuable in eco-design to reduce ecological damage by introducing arbosculpture along with topiary art, to become a part of a new direction in city design. *P. padus* was mentioned as a potential species in this context, capable of bearing crown formation by means of the flexibility of its branches [120]. However, *Prunus virginiana* (which may be confused with *P. padus*) is considered to be better for that purpose than *P. padus*, among other characters, because there were cultivars with purple leaves during the whole growing season in [121,122]. However, this species is an alien from North America and, similar to *P. padus*, is often considered a pest, as it is a host for the tent caterpillar (*Y. evonymellus*), which is a threat to other fruit plants like apple [7].

## 6. Discussion of the Reviewed Results

*Prunus padus* is hardy and the most northerly distributed of the genus *Prunus* in Europe and Asia. It is invasive, which make it valuable for securing slopes, preventing landslides. In its niche, *Prunus padus* competes with bushes and small and large trees, which are natural and ancient competitors, some of them since the last glaciation. The northerly distribution limits broadly coincide with the shore of the Arctic Ocean. Twelve widely distributed species, demonstrating the effective dispersal of pollen and seeds, were considered viable, among these *Prunus padus*, and have, as such, no particular conservation needs. The effective seed dispersal of the species, as inferred from post-glacial migration rates, may be partly responsible for its generally early post-glacial appearance, and may, in combination with the wide ranges and relatively large evolutionary potential, indicate that viable species are best able to cope with climatic change. In the Alps, it grows at a higher altitude than any other deciduous tree. Moreover, there is strong evidence that *P. padus* persisted in northern Europe during the last glacial maximum (approx. 20,000 BP), based on presence of the co-dependent leaf beetle (*Gonioctena internodia*) in northern Europe since the beginning of the last glaciation. It is therefore a viable plant that has grown in competition with other plants for a long time. It grows in niches generally in riparian woodland, not only near creeks and rivers but also on nearby hillsides and generally in woodland with humid soil, often together with alder (*Alnus incana*). The sub species *borealis* grows in montane or arctic communities in a lighter soil than ssp *padus.*

In addition of being a strong survivor, it is valuable for eco-design, and is also a good solitary park tree, with early flowering with a cloud of white flowers, that have a strong but not unpleasant smell. However, it may be strongly attacked by the small ermine moth (*Yponameuta evonymellus* L.). The caterpillars may weave giant webs over the whole tree, including the trunk, which has reduced its value as a park tree. However, today, when hopefully we are more concerned about nature and all its facets, it should be valued for demonstrating the effect of this pest, and the survival mechanism of *P. padus*. Disturbing forest communities, e.g., by frequent cutting of trees, is not beneficial for *P. padus*, and will make it difficult to establish niches in the forest. These cuttings are beneficial for the alien competitor *P. serotine* to establish. However, even though *P. serotina* establishes more frequently than European bird cherry in clear cuts, it will probably struggle to cross the ecotone between clear cuts and meadows into the forests. Anyway, *P. serotina* is in many ways similar to *P. padus* and manages well in European/Asian forests and has found its niche and will probably be a mild competitor to *P. padus* in the future. However, it will not be a threat to *P. Padus*, unless something dramatic happens (climate, diseases, pests) that changes the ecosystem in a direction more beneficial to one of the species.

Fruits, leaves and stem tissues are nowadays less used by farmers and rural population than some decades ago. However, people still collect berries to make syrup, jam and liquor. Moreover, the wood is attractive for wood carving and making cabinets, and has value as firewood. More importantly, it is concluded that bird cherry fruits, leaves and bark may be valuable sources of powerful natural

antioxidants for use in food, medicine, cosmetics and other fields currently processing antioxidants. The species should also be highlighted as a park tree in spite of being vulnerable for damage of the tent caterpillar. This pest and the ability of *P. padus* to survive the pest should rather be highlighted to demonstrate the magic of nature for the public. Moreover, it has value to secure slopes for landslide, that could be even more of a problem as climate change and so the rainfall and flooding.

These perspectives justify its ranking in category LC (least concern) in the IUCN Red List of threatened species [123]. However, the subspecies *borealis (syn. petraea)* is rare and locally threatened in some regions; therefore, national level monitoring is required. Germplasm collection and duplicated ex situ storage is also a priority for this sub-species [124].

## 7. Conclusions

*Prunus padus* is a species that has been present in Eurasia at least since the glacial maximum, approx. 20,000 years BP. It is the most northerly and at highest altitude growing of the *Prunus* species. It has an invasive character and is adapted to herbivory feeding and resistant to climate change. Human disturbance of riparian forests will make it harder to withhold the niches suited for *P. padus*. In spite of that, the species is adapted to herbivory, which cause more or less damage to the tree but not kill it; neither are alien trees such as *P. serotina* a serious threat. The strongest threat is probably frequent tree felling, which changes the succession of forest communities. Therefore, ssp. *padus* is not a threatened species; however, ssp. *borealis* is in many regions, and actions should be taken for its protection. Fruits, leaves and bark may be valuable sources of powerful natural antioxidants for use in food, medicine, cosmetics and other fields currently processing antioxidants. The species has value in hindering landslides and by introducing arbosculpture along with topiary art, to become a part of a new direction in city design. Tent caterpillars may weave giant webs over the whole tree, including the trunk, which has reduced its value as a park tree. However, today, when hopefully we are more concerned about nature and all its facets, it should be valued for demonstrating the effect of this pest, and the survival mechanism of *P. padus*. Cultivars with longer racemes and colored flowers have been developed, and could be valuable in gardens and parks.

**Funding:** This research was mainly funded by internal funds of the Norwegian Institute of Bioeconomy (NIBIO), section Horticulture, Ås. Some funding was provided by the author after his retirement, as time spent writing the article.

**Acknowledgments:** I would like to thank A.M Hietala, Section Fungal plant pathology in forestry, agriculture and horticulture (NIBIO) for reading and commenting the article, and help by retired Cand. Scient. Ecologiae Tor Bjørgen, Stjørdal municipality, Norway, for guidance through *Prunus padus* niches in Stjørdal municipality, Norway.

**Conflicts of Interest:** The author declares no conflict of interest.

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
