# Peer review of "The Status of Prunus padus L. (Bird Cherry) in Forest Communities throughout Europe and Asia"

_forests, doi:10.3390/f11050497_

Round 1
Reviewer 1 Report
I appreciate the hard work that went into this review.
I think with some rewriting of confusing areas the paper will be ready.
The wifi information is very suspect and my first response is to reject this paper completely but since it is a review you might not know about how pathologicial science is conducted.
good luck with the paper!
Author Response
I have removed the WI-FI info from the abstract, and changed the information around this in he article in such a way that the observation may be questioned
Reviewer 2 Report
“Prunus padus L. (bird cherry). Value as a multiple source for human use, but is its position threatened in forest communities throughout Europe and Asia? A review”
The aim of the submitted manuscript “is to give a status of the biological classification and phenology of P. padus, its position as a valuable tree in the forest community ecology and to manifest its ornamental and human health related value” (lines 47-49). The additional question was included it the title and concerns the threat in the forest communities within geographical range of the species. I understand that this statement works as an additional aim of the submitted review manuscript.
The manuscript is an interesting and valuable attempt to summary the information about ecology and human use of the species, however contains many shortcomings. The goals set to be accomplished are serious. This imply the need for deeper knowledge review of some aspects of the work and better organization of the whole text. The detailed suggestions are listed below.
a) Introduction
Lines 70-74 – Taxonomy of the species. As it is a new paragraph, it would be good to write what genus is this. There should be more information about taxonomy provided. For example, Prunus padus ssp. petraea (not “petraeae”) can be treated also as a synonym of Prunus padus ssp. borealis. It is important as at the end (lines 348-356) Author summarizes the threat to the species and subspecies.
Line 72 – Instead of using “nordherzynischen Bergland” it would be much better to write where it is, using for example names of mountain ranges. This name is not commonly used and understandable.
Line 75-76 - “The hardy species...” - the sentence is not well understandable to me. I do not understand what is “hardy” and to what belong “distribution limits” – this was used in a new paragraph.
b) community ecology – title of the chapter 3.1 “Pheonological influence” does not reflect its content well. This subchapter contain information about ecological niche, regeneration mechanisms, disturbance and succession, but also some aspects of forest management (lines 128-131). The name of the subchapter should be changed and would be good to move the “management practices” to the other place. I think, that for example this could be joined with “electromagnetic pollution” (lines 330-334), anthropogenic pollution (lines 336-340) and even restoration practices (lines 340-343) under a new subchapter treating about human disturbance.
Line 105 - “when it comes to ...” - it is not necessary to start the sentence in this way, it could be shortened
c) chapter 3.2. The herbivore influence.
This chapter contains three subchapters. I think, that it could be treated more broadly (also with a more broad title) and some not well organized parts from the further text moved here. In the introduction (lines 142-143) there could be more information placed. The parts of the following text could be moved to this chapter, even as an another subchapters:
- information from “other factors...” chapter (lines 317-329 information about other pests),
- information about beaver (lines 334-336) could be a bit deeper and moved here,
- information about wild boar (lines 343-347) – also treated in a bit deeper way.
Only two main herbivore insects were described here as the most dangerous to Prunus padus. I am not a specialist in this topic, but with a quick search in the available literature I have found the information about Gonioctena quinquepunctata which is known as a “serious leaf pest” of the species (for example according to Mąderek et al. 2015. Influence of native and alien Prunus species and light conditions on performance of the leaf beetle Gonioctena quinquepunctata. Entomologia Experimentalis et Applicata 155: 193-205 DOI: 10.1111/eea.12298 ). I do not want to force Author of the manuscript to cite this work, but I feel that some more valuable information in this subchapter could be suppelemented.
d) 3.3. chapter “Human interference on forest ecology” - the chapter is mainly about Prunus serotina, ecotones etc. maybe it would be better to move it close to the proposed chapter about “anthropogenic influence”?
Lines 260-263 – Climate change – would be good to treat as an other closely related subchapter and bit better described – for example in the context of species range and requirements.
e) 4. Forest community and ecosystems interaction
First and part of second paragraph (lines 265-300) is mainly about seed dispersal. It would be better to move this part closer to phenology or morphology (first parts of the text).
Second part (lines 300-315) is about the ecological niche and forest communities. Similar aspects are described within lines 105-117. The review on forest communities should be treated more deeply, include more information from other parts of the species range and be divided according to Prunus padus subspecies. For example, in central Europe Prunus padus is connected with alder-ash riparian forest and alder carrs, but Prunus padus ssp. petraea also occur in mountains (sometimes defined there as Padus petraea). There also should be available some more information about role of the species in ecological succession (in central Europe especially within riparian habitats). This is especially important due to the main aim of the work.
f) The possible threat to Prunus padus (lines 348-356).
According to the title it is an important part of the review. This should be an another chapter where the knowledge is summarised and some regional publications assessed, not only general Red List with a note of subspecies. I would see this after chapter 5 and being something like a discussion of the results of the conducted review. Thus it should include the summarised information about human value and amount of current usage.
g) According to the title of the manuscript, this chapter should be significantly expanded and ordered, especially subchapter 5.3. “other values”.
f) the references are not well formatted. Please use better source to cite than wikipedia (reference 2).
Pointing a very strong competition with Prunus serotina seems a bit strange to me. I agree that in some habitats these species can compete, but at least the same can happen with Alnus glutinosa and many more species (for example Salix cinerea, Cornus sanguinea, Rhamnus cathartica etc.), especially during secondary succession, or after the disturbance.
Author Response
lines 70-74: I have added the genus name and included that Prunus padus spp borealis is synonomus with Prunus padus spp borealis. Adding the ITIS catalogue as reference [15].
Line 72: I have changed the text according to the reviewer.
Line 75-76 I have tried to improve the text, to make it more precise, according to the reviewer.
I have made two sub.chapters under: 2. Botanical classification, 2.1 Classification and 2.2 Phenology and placed "Seed dispersal" under Phenology as recommended by the reviewer
b) Community ecology: I have created a new sub-chapter 4.2 Human interference and have made the changes recommended by the reviewer.
line 105: shortened according to reviewer
I have restructured Chapter 3, made the suggested movements and added a new sub-chapter 3.3.3: Injury caused by mammals.
Moved data about the beaver and the wild boar here.
e) Forest community and ecosystem interactions
f, g) I have done my best
Round 2
Reviewer 1 Report
Forests-767056
Review Prunus padus L. (bird cherry). Value as a multiple source for human use, but is its position threatened in forest communities throughout Europe and Asia? A review.
Rolf D. J. Nestby
General Comments: I have tried to figure out what changes were made regarding my questions and comments. I only see one comment back from the author that he removed the wi-fi reference from the abstract and left the wi-fi story in the text. Leaving this story in the paper is not acceptable since wi-fi damage has not been shown to be a real pathological fact. We can easily see what this type of mis information has done with the recent corona virus and destruction of cell phone towers in England.
I do not see any other responses to my questions and suggestions to improve the clarity of the sentences.
So I repeat these comments below- These need to be addressed or/and corrected- not ignored. Please list where and how you address these concerns by copying the text changes below the question in your response. Thank you.
You mention two insect problems- are they seen across the distribution of the species? Or just in Norway?
Line 15 You state the insects are in balance and all diseases also are in balance- where does this prefect harmony occur and how can you say this? Have there been survey’s and data collected across all of Asia and Europe to make these statements? I could believe you, but I don’t see the data presented in the paper. You can’t make statements like this without data.
Line 126-130 I do not see what these sentences do to further your story of where cherry’s place are in forest communities. If you are promoting the improvement of forest communities research - please put this in the discussion- These sentences are not needed here.
Line 87: How about – P. padus is an important species when it comes to carbon storage with an average of 47.6% of the biomass as carbon.
141-143 “Parallel to this, in grey alder (Alnus incana) stands the most important differentiating species in the Croatian association Equiseto hyemali- Alnetum incanae Moor 1958, is Prunus padus. This is a continental region along the watercourse of the river Drava. Here the grey alder forest occurs mainly in riparian and floodplain forests”.
Explain what differentiating species means. Maybe you could turn these three sentences into reverse order and weave in what you are trying to say about prunus.
145-150 - please explain what subsidies are? Is this potential food for fish from insects on the trees or nutrients from insect poop or something else ?
151-155 Sorry- maybe some readers love extra words and understand these terms but if you want a general science reader to understand- you need simpler words or ask someone to read the sentence and explain it in different words because I don’t know what you are saying. “In a study of structure mechanism of tree species diversity pattern in a near-mature forest in China, Prunus padus was at a 0-50 m scale, together with several other tree species classified among diversity accumulators [28”
Even with this following sentence that is supposed to explain the previous sentence I am lost with the words. “This implies that P. padus has positive facilitative interactions with other species, and that the target species would accumulate and maintain an over-representative proportion of diversity in its proximity [29
158-274 I have no issue with these paragraphs, but I have no idea where these insects are an issue in the world. Is this just in Norway or across the entire range of the species? It is hard to believe these two insects are the most important insect across the huge range of this tree. Giving examples of insect damage from some location is fine but I would not represent it occurs across the range- or maybe they do?? Please list where these insects are damaging in the distribution of the tree.
345 Not sure what this is talking about- sentence not complete. I do not know where this sentence is now.
Line 159-161 Create problems where? Norway- the entire range of the tree???
Line 474- 475 Please Remove sentence
Line 557- 559 Moreover, bird cherry (P. padus) is in balance with natural existing herbivores and thrives in the forests in spite of herbivore caused injuries. Electromagnetic pollution could be a problem at touristic places and in cities, but would not threaten the existence of P. padus in the large picture.
Do not repeat non science in the second sentence. Remove the sentence on electromagnetic pollution
The question still remains --how you can claim the tree is in balance?- I do not see you citing surveys or permanent plot data that shows this conclusion. You might claim that the tree is seen inhabiting forests where you live? But don’t claim the tree is in balance with its predators across its range as you do here.
Author Response
Hello,
It was not my intention to be ignorant. I appreciate that someone is willing to do review (I do some myself). However, the only comment from You as delivered from the editor the first time, was Your comment on WI-FI. I informed the editor about this, and expected that if there were more comments the editor would post them to me, but that dis not happen. It has made You frustrated and created some extra work for me. Anyway I got Your comments this second time, and I will certainly do necessary changes.
WI-FI: I have excluded this information from the article
Line 15: This is an abstract, but it is based on the information of several authors. However, the conclusion is mine, and since the bird cherry has existed since at least the last glaciation, I find it obvious that the species have "learned" to live together with its herbivores. I have added "troughout Europe and Asia".
Line 126-130: These lines are in chapter 2.2.1 Seed dispersal. I do not see why these sentences should not be here? (perhaps I am confused by the line nr?)
Line 87: Which in the latest version of the article is line 95! I accept Your suggestion.
Line141-143. I have explained differentiating in brackets
line 145-150: I have explained subsidies (not in detail)
!51-155: (163-168) I have tried to explain (in brackets the meaning of some special terms/words.
Line 158- 274. I have added som text in the start of the chapter. These two species are the main ones. However, regionally specific problems may arise.
Line 345: I is impossible for me to find out with line it is!
Line 159-16: I have added some text.
Line 557-559: I have removed the sentences You suggest. Thoug that the tre is in balance with its surroindings is quite cler. It is viable and has been there since the last glacial maximum, and it has an invasive character that help it to "protect" its habitat. So to be honest I think the sentence is OK. However, I give You the last word.
Thank You for a thoroughy review!
Reviewer 2 Report
“Prunus padus L. (bird cherry). Value as a multiple source for human use, but is its position threatened in forest communities throughout Europe and Asia? A review”
The second version of manuscript with the responses to submitted review is in my opinion still inadequately revised. Some of the text was rearranged, but the key things were not sorted out.
According to the aims and the title, the most important aspects are the analysis of threat and human use within whole geographical range of the species. The analysis of threat requires also at least a brief review of systematics of the species. This is due to the subspecies of Prunus padus are of the different threat categories. The threat should be review much more carefully and this was indicated in the first review of the manuscript.
For example:
Taxonomy
lines 70-74: some corrections have been applied but there is still a lack of the most basic informations. For example, Prunus padus ssp. petraea = ssp. borealis is present also in Carpathians (not only Western ones), nothing is written about other synonyms, sometimes still used in some countries (ie. Padus petraea).
The assessment of threat:
The is still a lack of most basic information, for example Prunus padus subsp. borealis – NT in Carpathians (“Carpathian Red List Of Forest Habitats And Species Carpathian List Of Invasive Alien Species. Draft Carpathian Red List Of Fish And Lamprey Species” Banská Bystrica: State Nature Conservancy Of The Slovak Republic, 2014). It is still only stated that “However, the subspecies borealis is rare and locally threatened in some countries; therefore, national level monitoring is required.” without citations (lines 479-480). “Carpathian” is rather regional than local.
Missing Carpathians in the review is misleading.
Citing Wikipedia
Reference 2 (indicated in first round of the review) was not corrected, Fig 2 is downloaded from Wikipedia and cited, but is based not on the Wikipedia but on the other published data (Chorological maps for the main European woody species, Data in Brief 2012, https://doi.org/10.1016/j.dib.2017.05.007).
… and some more indicated in the first round of the review.
In my opinion the main indicated aims of the review were not achieved.
Author Response
I have added some more information around borealis/petraea and its position in the Carpathian Mountains (It was not easy to find!)
lines 70-74: Synonyms - it has not been a goal in this article to give a full discussion of the taxonomy of Prunus padus.
The Wikipedia sitation is removed!